# Monitoring the Early Strength Development of Cement Mortar with Piezoelectric Transducers Based on Eigenfrequency Analysis Method

**DOI:** 10.3390/s22114248

**Published:** 2022-06-02

**Authors:** Guocheng Wang, Wenying Qiu, Dongkai Wang, Huimin Chen, Xiaohao Wang, Min Zhang

**Affiliations:** 1Tsinghua-Berkeley Shenzhen Institute, Tsinghua University, Shenzhen 518055, China; wgc19@mails.tsinghua.edu.cn (G.W.); qwy17@mails.tsinghua.edu.cn (W.Q.); wdk18@mails.tsinghua.edu.cn (D.W.); wang.xiaohao@sz.tsinghua.edu.cn (X.W.); 2Shenzhen International Graduate School, Tsinghua University, Shenzhen 518055, China; chenhm19@mails.tsinghua.edu.cn

**Keywords:** cement strength, structural health monitoring, piezoelectric transducers, smart aggregates, eigenfrequency

## Abstract

Monitoring the early strength formation process of cement is of great importance for structural construction management and safety. In this study, we investigated the relationship between the eigenfrequency and the early strength development of cement mortar. Embedded piezoceramic-based smart aggregates recorded the early strength of cement mortar. An eigenfrequency analysis model demonstrated the relationship between strength and frequency. Experiments were performed by using piezoelectric transducers to monitor the early strength formation process during the testing period. Three types of specimens with different strength grades were tested, and the early strength formation processes were recorded. The experimental results demonstrate that cement mortar strength has a good linear relationship with the resonance frequency, and the average square of the correlation coefficient is greater than 0.98. The results show that structural health monitoring technology is a feasible method of assessing structural safety conditions and has a broad market in the structural construction industry.

## 1. Introduction

Cement is widely used in buildings, bridges, dams, and other buildings, as a civil engineering material [1,2,3,4,5]. As an inorganic cementitious material in powder form, cement is also an essential raw material for making concrete. Cement is mixed with water to form a slurry which hardens in air or water, binding the various aggregates together. Commonly used cement is Portland cement, including ordinary Portland cement, slag Portland cement, volcanic ash Portland cement, fly ash Portland cement, etc. [6,7,8,9,10,11,12,13,14]. The strength of cement is gradually formed during hydration, and the final strength grade of cement is an essential factor affecting building strength. Monitoring cement strength is useful when studying the formation of building strength and ensuring the safety of building construction. The cement strength is related to maturity, which is the index of evaluating cement and its mixture formation and depends on temperature and time [15]. A cement mortar block is a test block made of mixed and hardened sand, cement, and water, with a standard proportion, often used to test cement strength [16,17,18]. As a standard test block with a fixed size, the study of its modal and eigenfrequency changes does have specific significance to cement strength formation.

Structural health monitoring (SHM) and damage identification are technologies that have received much attention in recent years. It is an important driving force in civil engineering construction to develop informatization and intellectualization. By using sensing technology to monitor the mechanical properties of a building structure, SHM can diagnose the damage, aging, and corrosion degree of the structure and evaluate its safety, durability, applicability, and bearing capacity in real-time [19,20]. Moreover, SHM technology is a way to minimize manual intervention in the building structure with automatic, real-time, continuous online monitoring and to make reliable estimates of the health of the building. It is a viable method to acquire and process data from a structure in an operational state and assess the key performance indicators of the structure with a nondestructive method [21]. There are many sensors used for SHM, including cement-based sensors [22,23,24,25,26], fiber optic sensors [27,28,29,30], and piezoelectric ceramic sensors. Compared with other methods, piezoceramic has some unique advantages based on the characteristics of piezoelectric materials, which can be used not only as sensors but also as actuators to provide vibrational excitation for active SHM.

The basic principle of the piezoelectric wave propagation method is to attach a pair of piezoceramics, called smart aggregates (SA), to the surface of the structure or to embed them into the structure. One piezoelectric transducer transmits stress waves and the other receives stress waves. By analyzing the changes between the transmitted and received signals, such as the attenuation of amplitude and the delay of the propagation time caused by strength formation, the mechanical properties and damage condition of the structure can be identified. When the structure is significant, embedded piezoelectric transducers are better suited to detecting internal structure factors of the concrete structure than surface-bonded transducers. In addition, this approach can overcome the disadvantage of lead zirconate titanate (PZT) vulnerability and provide protection for the core of PZTs for better structural health monitoring. To provide electrical safety and mechanical security, respectively, for the PZT patches, a waterproof layer and protective layer were used to wrap them [31].

Piezoelectric SA is a piezoelectric piece or piezoelectric functional element embedded into a structure in various ways. Compared with external paste transducers, this mounting method will not affect structure shape, thus reducing external interference factors and improving the reliability of the test results. However, the stability and reliability of the structure could be affected by an embedded piezoelectric element to a certain degree. This method can effectively extend the longevity of piezoelectric material and provide long-term, dynamic and real-time health monitoring and damage identification. In 2006, Song et al. [32] used an innovative piezoelectric-based approach to conduct early-age strength monitoring of concrete structures. Piezoelectric transducers in the form of SA were embedded into concrete structures as actuators and sensors during casting for strength monitoring purposes. In 2012, Hou et al. [33] used a PZT-based SA for seismic compressive stress measurement. The sensing system for dynamic compressive stress included proposed SAs and a commercially available charge amplifier. With a lower limit of frequency response stress amplitude, the sensing system is also capable of monitoring seismic pressures of low- and middle-rise buildings subject to moderate earthquakes. Meng et al. [34] achieved SHM of reinforced concrete columns under a static loading procedure by using PZT smart aggregates as sensors and actuators based on an active sensing system. In 2014, a combination of SA for monitoring the overall structural condition and surface-bonded PZT transducers for monitoring the crucial sections of construction was proposed as an effective SHM method for large infrastructures [35]. In 2017, Kong et al. [36,37] finished developing a novel embeddable spherical smart aggregate (SSA) for health monitoring of concrete structures. This novel SA offers omnidirectional actuating and sensing capabilities. In 2019, Zou et al. [38] provided a comprehensive examination regarding temperature effects on SA-based active monitoring methods for concrete structures. Zhou et al. [39] used a series of piezoceramic transducers that enabled the time-reversal method to identify the structural damage mechanism of basalt fiber reinforced plastic (BFRP) bar-reinforced concrete beams. Four SA transducers were bonded to the surface of a concrete beam to detect its damage in the bend-shear zone, the pure bending zone, and the overall zone. Gao et al. [40] made a novel tubular smart aggregate designed to be more compatible with 2-D structures. As seen by the above research, piezoelectric smart aggregates have been widely used in the research and application of structural health monitoring. However, their potential for strength monitoring of specific shapes and sizes, as well as the change of cement strength during curing by eigenfrequency analysis, has not yet been discovered.

In this paper, a novel method of monitoring the early strength development of cement mortars using embedded SA is proposed. Cement mortars were cast as standard specimens and analyzed by eigenfrequency analysis. The change of cement strength affected the maximum amplitude and resonant frequency of the received signal through wave propagation. The experimental results suggest that the actual strength of the cement can be reflected in the wave propagation characteristics. It proves that eigenfrequency is correlated with the strength of cement through maturity, and an exact functional connection was established.

## 2. Wave Propagation Monitoring and Eigenfrequency Analysis Method

The piezoelectric wave propagation method uses a pair of SAs embedded in the structure. One SA is used for transmitting stress waves and the other serves as the receiver, as shown in Figure 1. Stress wave parameters such as frequency, amplitude, and waveform, are directly related to the mechanical properties of the monitored structure, including the Young’s modulus, Poisson’s ratio, and eigenfrequency. When the stress wave propagates, it carries real-time information about the changes in the mechanical parameters of the structural material. The structural strength is identified by analyzing the signals received by the sensor under different conditions, such as the attenuation of the signal amplitude, the change of mode, and the delay of the propagation time caused by the strength. This method has a particular inhibitory effect on environmental noise and interference, which ensures the accuracy and reliability of the monitoring process and results.

The eigenfrequency or natural frequency is the discrete frequency at which a system tends to vibrate. Resonance occurs when the frequency of the external force is the same as the eigenfrequency of the system. In a resonant state, the vibration amplitude of the system will be maximized and will be significantly different from that in a non-resonant state. Based on the self-excitation characteristic of SA, the external force of frequency change can be applied to the system by applying a sweep signal to SA. The eigenfrequency of the system can be found by analyzing changes in the received signal.

The change of signal is related to the shift of structure eigenfrequency. Eigenfrequency is often related to the stiffness and mass of the structure itself [41,42,43,44]. In theory, the Young’s modulus changes regularly during the early strength development of concrete or cement mortar [32]. Structure stiffness and the Young’s modulus are equivalent to some extent, so the eigenfrequency analysis method could indicate the early strength formation process of cement mortar.

## 3. Experimental Method

### 3.1. Design and Fabrication of Smart Aggregates

Piezoceramic patches (PZT-5H, Baoding Hongsheng Acoustic Electronic Equipment Co., Ltd., Baoding, Hebei, China) were selected according to piezoelectric response coefficient and mechanical properties. As embedded transducers in concrete and cement mortar, the strength of piezoelectric ceramics should not be lower than the high strength and stiffness of concrete and cement mortar after forming. Additionally, the embedded transducers must have certain dimensional compatibility. The specific indicators are given in Table 1. SA has dual functions of excitation and reception as both the actuator and the sensor. Moreover, considering the sensitivity under the action of transmitting power and high frequency, PZT-5H was selected.

As shown in Figure 2, the SA consisted of a PZT patch with dimensions of 10 mm × 10 mm × 0.3 mm and an epoxy resin waterproof layer of 12 mm × 12 mm × 1 mm in the center of a cubic cement block of 20 mm × 20 mm × 20 mm with a protective layer of similar dimensions. A multimeter measured the resistance of each PZT patch before welding the wire to it. The infinite proved the whole monitoring system was packaged well, and the signal was transmitted through the stress wave instead of the current in the concrete. After the PZT patch was welded, a layer of epoxy resin potting adhesive was evenly applied to its surface using a polyvinyl alcohol (PVA) mold. After curing for 24 h, the epoxy resin-covered PZT patch (Figure 3a) was demolded by dissolving the PVA mold in water. The rigidity of cured epoxy resin is similar to that of cement mortar specimens, hence it had little influence on the regular operation of the PZT transducers, ensuring exemplary acoustic coupling. In addition, epoxy resin avoids short circuits while using PZT patches and direct contact between PZT patches and cement mortar as a waterproof layer.

The epoxy resin-covered PZT patch was then put into a new PLA mold, and the mixed cement mortar was cast into the mold (Figure 3b). To avoid errors caused by the difference in material properties, the same cement material as the specimens was used. The cast cement mortar was vibrated using a vibratory table and then compacted and smoothed. Finally, demolding was carried out after 48 h of curing, and the completed SA is shown in Figure 3c. SAs were used for early strength development monitoring of cement mortar specimens after 14 days. The mass proportion in cement mortar specimens, subsequent hydration heat, and strength variation of the cement mortar protective layer are tiny, hence the early strength development monitoring of cement mortar specimens was affected very little. The cement mortar protective layer of the SA had the same mechanical properties as the monitored cement mortar concrete to ensure the exemplary mechanical coupling and interface coupling of SA and monitoring structure. The cement mortar protective layer also bore a certain load to protect the PZT patch. The mass proportion of the SA was tiny and the mechanical properties of the SA are similar to the cement mortar specimens. No other sensors or impurities were between the two SAs for monitoring. Therefore, embedded SAs monitored the strength of cement mortar without affecting the strength development of cement mortar specimens.

### 3.2. Preparation of the Cement Mortar Test Specimen

Three types of cement, i.e., Portland slag cement 32.5 (P.S 32.5), ordinary Portland cement 42.5 (P.O 42.5), and ordinary Portland cement 52.5 (P.O 52.5), were selected for cement mortar specimens of 40 mm × 40 mm × 160 mm. The SAs were fixed at both ends of the specimen mold at a distance of 80 mm before casting and each SA was 40 mm away from one end of the cement mortar specimen. As shown in Figure 4a, three groups of specimens (S1, S2, and S3) were fabricated with different types of cement and ingredients (Table 2). The primary raw materials of ordinary Portland cement are silicate, gypsum, and some mixed materials. In addition to gypsum and silicate, Portland slag cement contains granulated blast furnace slag and other mineral materials. Because the composition of ordinary Portland cement and Portland slag cement is different, the hydration of ordinary Portland cement is faster than Portland slag cement. In terms of particle size, ordinary Portland cement is finer than Portland slag cement, so the mixture will be more uniform. Cement mortar specimens harden gradually at 20 °C curing temperature and 90% curing humidity after they are fabricated. The process is called setting or hardening. Fresh cement mortar or concrete gains strength most rapidly during the first few days. The strength of cement mortar or concrete will become stable after 28 days. The design of cement mortar or concrete structures is generally based on the 28th day strength. The strength of cement mortar or concrete before 28 days is called early strength.

### 3.3. Test Procedure

Due to the high stiffness of cement mortar specimens and the high driving voltage requirement of the PZT patch, a 100-volt swept sine signal from 10 Hz to 200 kHz was used to actuate the SA and generate stress waves. The specimens were continuously monitored for 28 days until the cement strength was fully formed and there was no significant change. The experimental monitoring system is shown in Figure 4b. The whole system included four parts: the signal generation module with the signal generator (DG1062Z, RIGOL, Beijing, China); the amplifier module with the voltage amplifier (ATA-2082, Agitek, Xi’an, Shaanxi, China); the SAs; the oscilloscope (TBS1102, Tektronix, Beaverton, OR, USA). The schematic diagram is shown in Figure 4c. The SA as the actuator was connected to the amplifier and the signal generator generated a sweep signal as shown in Appendix A. The other SA, as the sensor, received the stress wave that the actuator transmitted and displayed the waveform on the oscilloscope. Three extra cement mortar specimens for each cement type without embedded SAs were also prepared and tested using a destructive compression test on the 3rd, 5th, 7th, 14th, and 28th days for comparison.

## 4. Results and Discussion

The characteristics of a resonance signal and the change rule with time can be revealed by analyzing the received signal under a frequency sweep excitation signal. The change in cement strength is most dramatic during the first few days of hydration, and thus the 3rd, 5th, and 7th days were selected as representative time points, in addition to the intermediate and final time points of the strength formation process. The received signal was filtered by FFT to filter out 50 Hz power frequency noise. After the received signal was denoised, the received signals in a sweep cycle on the 3rd, 5th, 7th, 14th, and 28th days are shown in Figure 5, Appendix A. In the finite element simulation (made by COMSOL Multiphysics, and the detailed setting parameters are given in Appendix A), the first eigenfrequencies of P.S 32.5, P.O 42.5 and P.O 52.5 cement mortar specimens were 4738 Hz, 5882 Hz, and 6741 Hz. The second eigenfrequencies were 10,913 Hz, 11,343 Hz, and 12,875 Hz, respectively. In a sweep period signal of experiment, the received signal also has two apparent peak values in the early stage. The first peak value is about 5 kHz, which could be related to the first eigenfrequency of the specimen, and the second peak value is about 10 kHz. The amplitude of the signal from 20 kHz to 100 kHz is relatively stable, which could be caused by the gradual departure from the eigenfrequency of the specimen. The amplitude of a signal with a frequency greater than 100 kHz is minimal, which could be caused by the excitation signal frequency far exceeding the eigenfrequency of the specimen. The signal under sweep excitation also had obvious changes during the strength forming process of the specimen for 28 days. The received signal intensity under the frequency sweep excitation gradually decreased with time after the specimen reached the initial forming. The peak value of the received signal under the sweep excitation also drifted slowly to the higher frequency.

In a system of a single degree of freedom without damping, the natural angular frequency is defined as:(1)$$\omega_{0}{=2\pi f}_{0}=\sqrt{k/m}$$
where *ω*_0_ is the natural angular frequency with unit rad/s, *f*_0_ is the natural frequency, *k* is the system’s stiffness, and *m* is the mass. The stiffness is proportional to the Young’s modulus so that the natural frequency can be defined as:(2)$$f_{0}\;\propto \;\sqrt{E/m}$$
where *E* is the Young‘s modulus. Furthermore, the Young’s modulus *E* is linearly related to the concrete strength Sm. Therefore, the natural frequency *f*_0_ is correlated with the concrete strength *S_m_* through the status of the Young’s modulus [32]. Assuming mass is constant, the relationship between frequency and strength can be defined as:(3)$$f_{0}\;\propto \;\sqrt{S_{m}}$$

The pivotal factors affecting the early strength of concrete or cement mortar are temperature and time, as the raw material and composition ratio are determined. Maturity is a physical quantity combining temperature and time. The relationship is based on the Saul-Bergstrom function (Equation (4)) [45,46].
(4)$$M_{s}=\sum^{\;}\left(T_{i}+10\right)t_{i}$$
where *M_s_* is the maturity with unit °C∙d, *T_i_* is the curing temperature, and *t_i_* is the curing time under *T_i_* temperature. In consideration of constant temperature during the curing process of cement mortar block, the function is simplified as:(5)$$M_{s}\;\propto \;t$$
where *t* is the curing time of concrete or cement mortar.

There is a certain functional relationship between strength *S_m_* and maturity *M_s_* [47,48,49]. The functional relationship between strength *S_m_* and maturity *M_s_* is similar to the power function through experimental verification and regression. Because of the linear relationship between maturity and time, so the strength can be expressed as:(6)$$S_{m}\;\propto \;t^{r}$$
where *r* is exponential coefficient.

With Equations (3) and (6), the function of time and the natural frequency can be defined as:(7)$$f_{0}=a\times t^{b}$$
where *a* and *b* are coefficients.

The resonant frequencies of P.S 32.5, P.O 42.5, and P.O 52.5, obtained by swept signals are shown respectively in Figure 6a–c. After data fitting, the curves of each specimen’s first and second resonant frequencies against the curing time illustrate the change of eigenfrequency over curing time. The variation in eigenfrequency with time conforms to the rule of the power function. The square of the correlation coefficients (*R*^2^) of all fitted curves is greater than 0.9 and the absolute value of the correlation coefficients (|*R*|) is greater than 0.95. According to the correlation coefficients, the fitting results are highly correlated. In the comparison of three different cement specimens, the coefficient *a* of the fitting curve also gradually increases with the increase in cement strength grade. The reason is that the specimens with higher cement grades have higher initial strength in the hydration process.

The finite element model of cement mortar test block embedded with SAs is established by COMSOL Multiphysics. In the simulation, the contact interface between different materials is a fixed constraint. The detailed setting parameters of materials are given in Appendix A. The simulation of the cement mortar modal displacement with certain eigenfrequency is normalized and shown in Figure 7. In Figure 7a, the mode at the first eigenfrequency is mainly bending deformation. The position of PZT patches is just at the neutral surface of specimen bending, and the extrusion and tensile deformation are small. In Figure 7b, the mode at the second eigenfrequency is mainly extrusion and tensile deformation. The deformation direction of the second eigenfrequency is consistent with the propagation direction of the stress wave produced by the PZT patch. Therefore, the signal acquisition intensity of the sensor at the working frequency close to the second eigenfrequency is better than that of the first eigenfrequency. This finite element simulation is also in accord with actual experimental results. The amplitude of the receiving signal with a 10 kHz excitation signal (nearly 2nd resonant frequency) is higher than others with different frequency excitation signals.

The cement mortar compressive strength value during curing time is shown in Figure 8a. The compressive strength test of cement mortar specimens was carried out simultaneously with the monitoring by SAs. Each type of cement was tested for compressive strength on days 3, 5, 7, 14, and 28. The curve fitted with the data shows the actual change of cement strength during curing time. The compressive strength test results were compared with the curve of the resonant frequency of the specimens obtained from the SAs with curing time, and it was found that the two are highly similar.

The relationship between daily strength and eigenfrequency of cement mortar in the hydration process was formulated and the linear fittingwas made as shown in Figure 8b–d. The following relationship can be obtained:(8)$$f_{0}=c+d\times S_{m}$$
where *c* and *d* are coefficients. The *R*^2^ of all fitted curves is greater than 0.98. The fitting degree of the first resonance frequency is slightly better than that of the second. With the increase in cement strength grade, the initial strength of the specimen becomes higher, and the initial resonance frequency also increases. The sensor sensitivity, i.e., the slope of the fitting line, is almost the same for the two types of ordinary Portland cement. The sensitivity of Portland slag cement is nearly twice that of ordinary Portland cement, perhaps due to the material characteristics of different types of cement. The analysis method of eigenfrequency is often used in the nondestructive monitoring of structures, which plays a significant role in monitoring the micro change of structures [42,43,44]. Structural damage and stiffness change both affect the eigenfrequency change. According to the change of characteristic frequency detected by SA, the change in cement mortar strength can be predicted based on experimental analysis.

## 5. Conclusions

In this paper, piezoelectric SA was used to monitor the early strength formation of cement mortar specimens with different strengths of cement. Excitation signals given to specimens were from 10 Hz to 200 kHz. The emitted stress wave resonated with the cement mortar specimen at the same frequency point as the specimen eigenfrequency. SA also detected the frequency of resonance as a sensor. The relationship between the strength changes and the eigenfrequency was summarized by analyzing the eigenfrequency and comparing the variation of received signals. There is a reasonable linear relationship between strengths and resonance frequencies of cement mortar with *R*^2^ of all fitted curves greater than 0.98. Consequently, the method of describing the early strength development of cement mortar by SA monitoring proves to be feasible. Likewise, the finite element simulation and compressive test were performed for verifying the experimental results. The proposed embedded piezoelectric transducers method with eigenfrequency analysis has the potential to monitor the early strength formation of cement. In the future, some improvements could be made regarding the sensitivity and accuracy of the device.

## Figures and Tables

**Figure 1 sensors-22-04248-f001:**
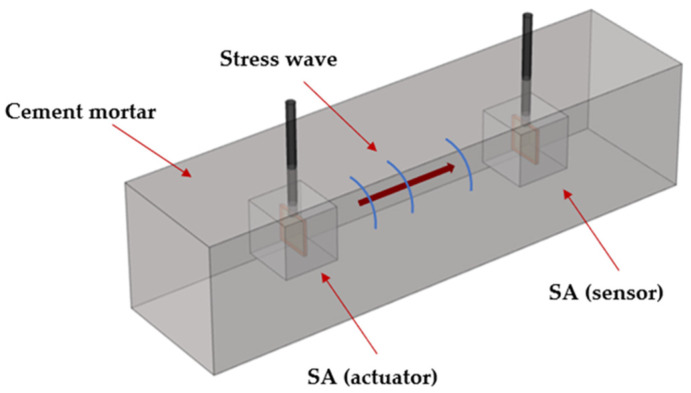
The configuration of the piezoelectric sensing system in the cement mortar test block.

**Figure 2 sensors-22-04248-f002:**
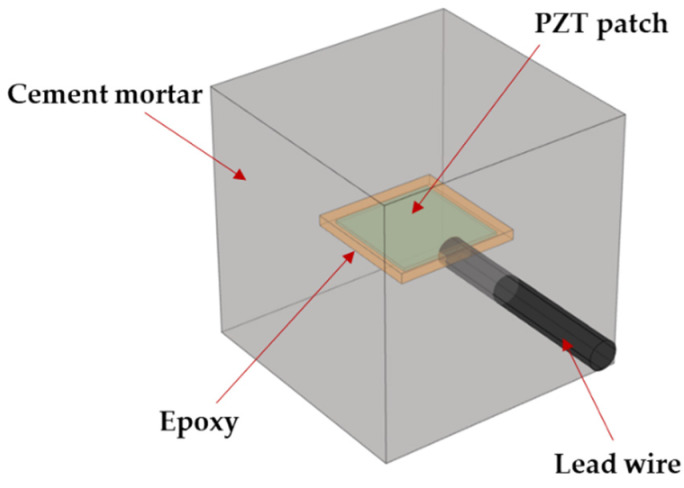
The structure of smart aggregate.

**Figure 3 sensors-22-04248-f003:**
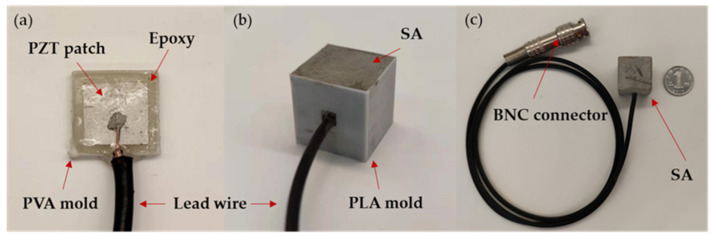
(**a**) PZT patch covered with an epoxy resin layer in a PVA mold; (**b**) Cement mortar with the epoxy resin-wrapped PZT patch cast in a PLA mold; (**c**) A photo of a smart aggregate.

**Figure 4 sensors-22-04248-f004:**
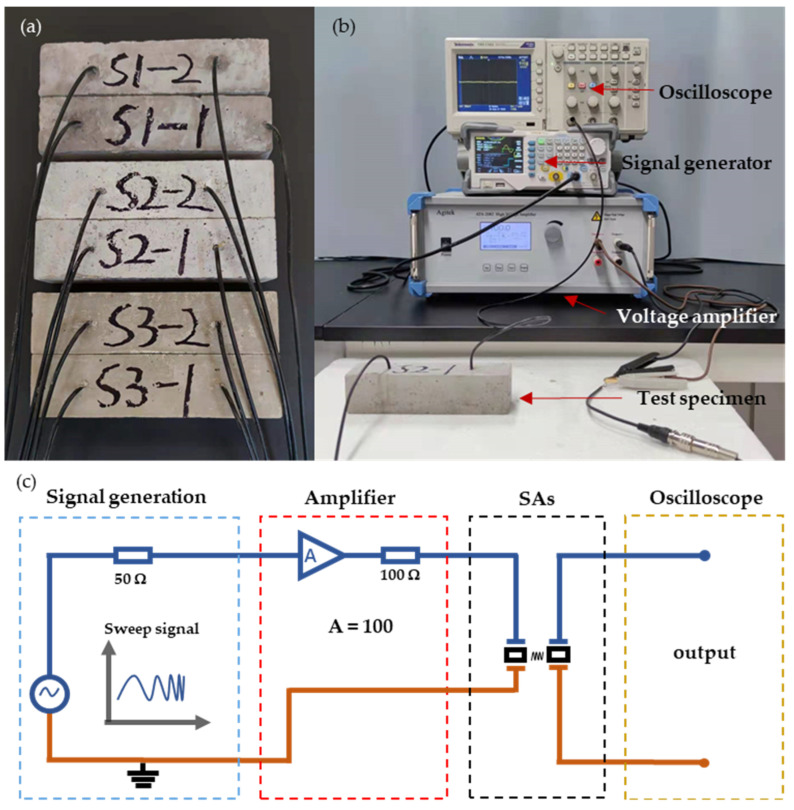
(**a**) Cement mortar specimens with embedded SAs; (**b**) Experimental monitoring system; (**c**) Schematic diagram of the monitoring system with four unit elements: the signal generation module, amplifier module, SAs, and oscilloscope.

**Figure 5 sensors-22-04248-f005:**
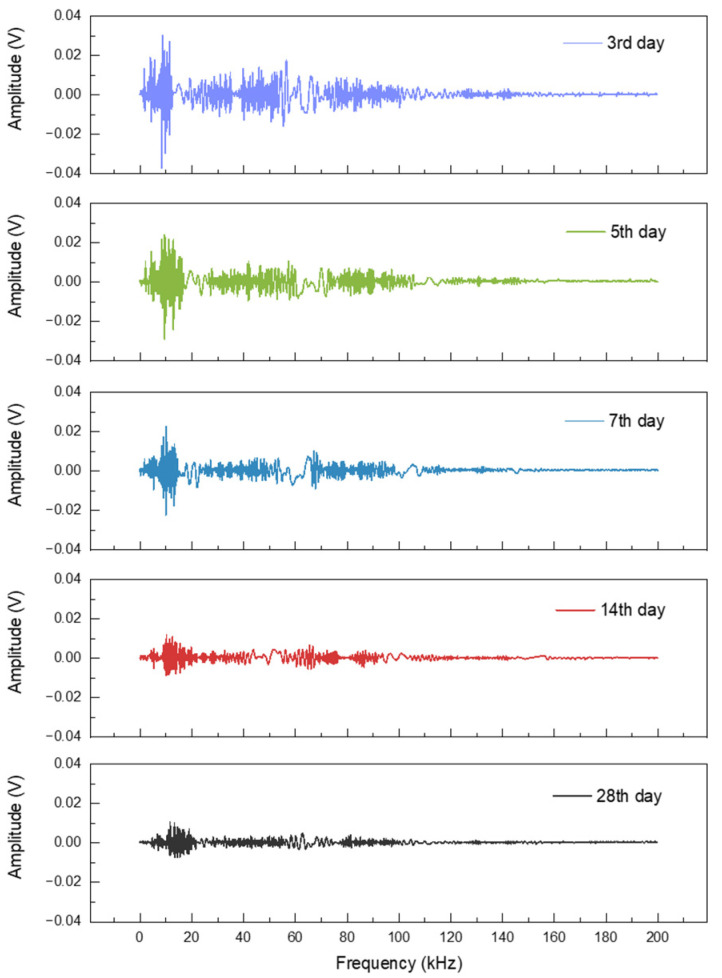
The received signal of P.O 42.5 specimens with swept sine excitation on the 3rd, 5th, 7th, 14th, and 28th days.

**Figure 6 sensors-22-04248-f006:**
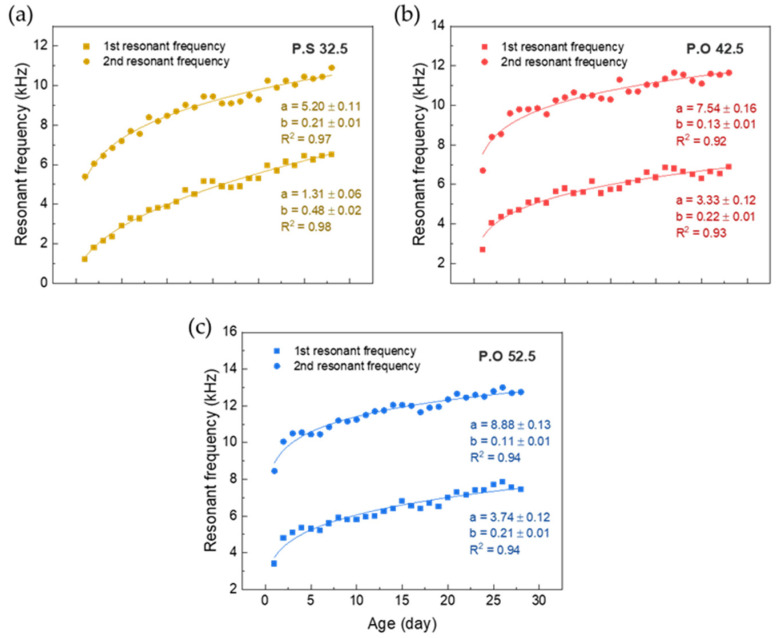
Experimental resonant frequency: (**a**) Resonant frequency of P.S 32.5 variation during time; (**b**) Resonant frequency of P.O 42.5 variation during time; (**c**) Resonant frequency of P.O 52.5 variation during the time.

**Figure 7 sensors-22-04248-f007:**
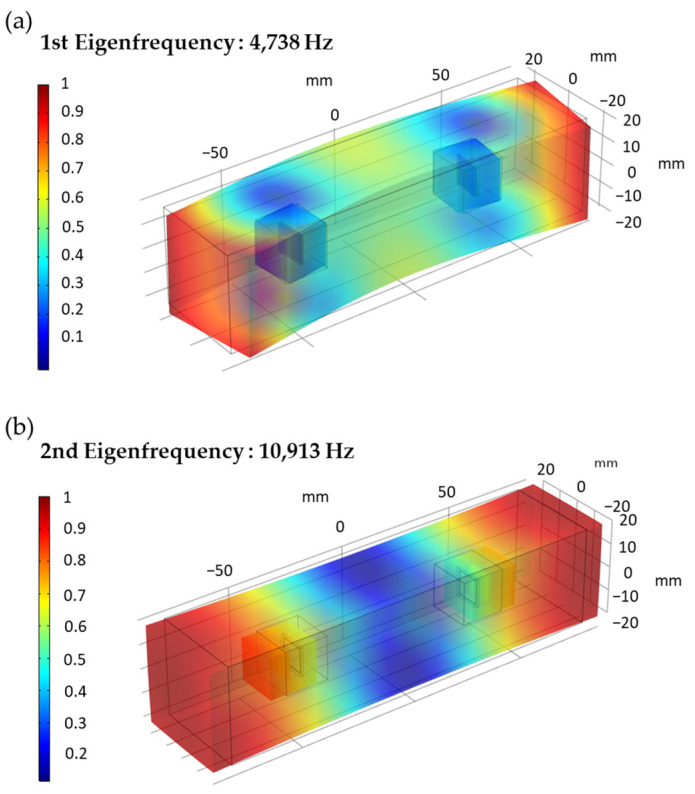
Finite element simulation of specimen eigenfrequency: (**a**) Modal displacement with 1st eigenfrequency of P.S 32.5; (**b**) Modal displacement with 2nd eigenfrequency of P.S 32.5.

**Figure 8 sensors-22-04248-f008:**
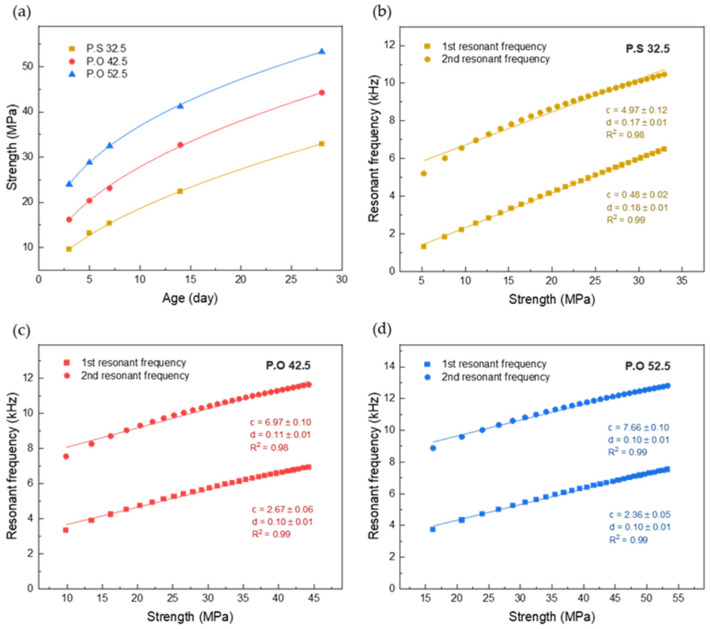
The cement mortar compressive strength and eigenfrequency value during curing time: (**a**) The variation of cement mortar compressive strength over time; (**b**) The variation of P.S 32.5 cement resonant frequency with strength; (**c**) The variation of P.O 42.5 cement resonant frequency with strength; (**d**) The variation of P.O 52.5 cement resonant frequency with strength.

**Table 1 sensors-22-04248-t001:** Performance parameters of PZT-5H.

Parameter	Numerical Value	Unit
Density	7.45	10^3^ kg/m^3^
Young’s modulus	46	GPa
Mechanical quality factor	70	/
Poisson ratio	0.33	/
Planar electromechanical coupling factor	0.65	/
d_31_, d_32_	−186	10^−12^ C/N
d_33_	670	10^−12^ C/N

**Table 2 sensors-22-04248-t002:** Ingredient of the cement mortar specimens.

Specimens	Cement Type	Cement Content (g)	Sand (g)	Water (g)
S1	Portland slag cement 32.5	300	900	150
S2	Portland cement 42.5			
S3	Portland cement 52.5			

## Data Availability

The data presented in this study are available on request from the corresponding author.

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
