# Peer review of "Monitoring the Early Strength Development of Cement Mortar with Piezoelectric Transducers Based on Eigenfrequency Analysis Method"

_sensors, 2022, doi:10.3390/s22114248_

Round 1

Reviewer 1 Report

In this paper, eigenfrequency analysis method was used to monitor the early strength development of cement mortar. The results showed a good linear relationship between cement mortar strength and resonance frequency that demonstrate this method has potential to monitor the early strength formation of cement. However, beside some issues need to be addressed, the main question is: what is the longest propagation distance of the stress wave in cement mortar? Does the result affect by the change of distance between the actuator and the sensor?

Detailed issues:

1) The sentence in line 135-136 is repeat with line 128-129, it should be deleted.

2) line 138, please give the detailed theory or related references.

3) Line 150, what does “reflection” mean? Does term “excite” is more appropriate here.

4) Section 3.2, please give the curing conditions of cement mortar, i.e., temperature and humidity, etc.

5) Section 3.3, please give the waveform of excite signal.

6) Fig.4a, the position of lead wire (S1-1) is different with Fig.1, does the change of SA location affect the test results.

7) Line 233, please give the detailed denoise method.

8) Line 235, there are many apparent peak values in the early stage from Fig.5, why choose 5 and 10 kHz as the analysis frequency. In addition, please give the first eigenfrequency of the specimen.

9) Line 291, the reviewer did not find the Table S1 in SI.

10) Please give the fit formula of the strength vs. age in Fig.8a, and clarify its accuracy.

Author Response

Dear Reviewer,

Thanks  for  your recognition of our paper and valuable comments. We have revised the manuscript according to your comments.

Sincerely, 

Min Zhang

5/30/2022

Reviewer 2 Report

Dear Authors,

thank you for that manuscript: very interesting idea, well written, good results = impressive!

just one question: how about the risk of mortar failure due to inserted piezos?

Author Response

Dear Reviewer,

Thanks  for  your recognition of our paper and valuable comments. We have answered your question in the attachment.

Sincerely, 

Min Zhang

5/30/2022

Round 2

Reviewer 1 Report

The reviewer think this manuscript can be accepted.